# Distinguished Quantized Guidance for Diffusion-based Sequence Recommendation

## Abstract

Diffusion models (DMs) have emerged as promising approaches for sequential recommendation due to their strong ability to model data distributions and generate high-quality items. Existing work typically adds noise to the next item and progressively denoises it guided by the user's interaction sequence, generating items that closely align with user interests. However, we identify two key issues in this paradigm. First, the sequences are often heterogeneous in length and content, exhibiting noise due to stochastic user behaviors. Using such sequences as guidance may hinder DMs from accurately understanding user interests. Second, DMs are prone to data bias and tend to generate only the popular items that dominate the training dataset, thus failing to meet the personalized needs of different users. To address these issues, we propose Distinguished Quantized Guidance for Diffusion-based Sequence Recommendation (DiQDiff), which aims to extract robust guidance to understand user interests and generate distinguished items for personalized user interests within DMs. To extract robust guidance, DiQDiff introduces Semantic Vector Quantization (SVQ) to quantize sequences into semantic vectors (*e.g.*, collaborative signals and category interests) using a codebook, which can enrich the guidance to better understand user interests. To generate distinguished items, DiQDiff personalizes the generation through Contrastive Discrepancy Maximization (CDM), which maximizes the distance between denoising trajectories using contrastive loss to prevent biased generation for different users. Extensive experiments are conducted to compare DiQDiff with multiple baseline models across four widely-used datasets. The superior recommendation performance of DiQDiff against leading approaches demonstrates its effectiveness in sequential recommendation tasks.

## CCS Concepts

• **Information systems → Recommender systems**.

## Keywords

Diffusion model, recommender system, vector quantization

**ACM Reference Format:**
Anonymous Author(s). 2018. Distinguished Quantized Guidance for Diffusion-based Sequence Recommendation. In *Proceedings of Make sure to enter the correct conference title from your rights confirmation emai (Conference acronym 'XX)*. ACM, New York, NY, USA, 9 pages. https://doi.org/XXXXXXX.XXXXXXX

## 1 Introduction

Sequential recommendation [15, 36, 41] focuses on capturing user interests through their historical interaction sequences to predict the next item with which the user will interact. Unlike traditional discriminative recommenders such as GRU4Rec [9], LSTM4Rec [47], and SASRec [15]) that aim to score and rank items, generative recommenders [18, 27, 33, 34] have emerged as promising alternatives, which emphasize the importance of item distribution modeling and generate the next item with generative models, such as GANs [6], VAEs [30] and, diffusion models [3]. Among these options, diffusion models (DMs) have recently gained attention in sequential recommendation [18, 22, 24, 44], due to their strong training stability and generation quality. Specifically, by progressively introducing noise to the ground-truth next-item representation and then gradually removing the noise guided by the user's interaction sequence, DMs learn to model the next item's distribution and have shown great potential in generating items that closely align with user interests.

When adapting diffusion models to sequential recommendation tasks, especially as item generators, there are two essential questions to answer: 1) How to extract accurate and robust *guidance* information for diffusion? And 2) how to effectively generate personalized item recommendations with the provided guidance. Despite the considerable success, DMs also introduce new challenges while answering the aforementioned questions:

- **Heterogeneous and Noisy Guidance:** The guidance aims to encode user interests based on the given historical interaction sequences, so that it could serve as a personalized condition [46] and enhance the accuracy of the subsequent item generation process. However, user interaction sequences in recommendation tasks are typically **heterogeneous** in lengths and contents [16, 17]. For example, a low-activity user may sparse interaction history records in recent days (and may only have one interacted item in extreme cases), then the guidance encoding may no longer provide sufficient information for the diffusion process. Even for users with longer history sequences, the interaction of items may contain noisy signals [8] due to stochastic user behavior (*e.g.*, misclick [19]). As illustrated in Figure 1, with the existence of noisy and sparse user sequences, the corresponding sequence encoding is susceptible to ambiguity. This may impede the model from accurately capturing user interests and consequently hinder the following generation process from exploiting this information.

- **Biased Generation:** Given the obtained guidance information, diffusion models estimate the added noise and remove it gradually [10]. However, the denoising process that generates items is prone to mode collapse and similar generation issues [12], especially when **biases** occur in input data [26, 31]. For example, some popular items may appear in a large portion of the data, which will receive sufficient training and precise generation, but may potentially overwhelm the learning of underrepresented

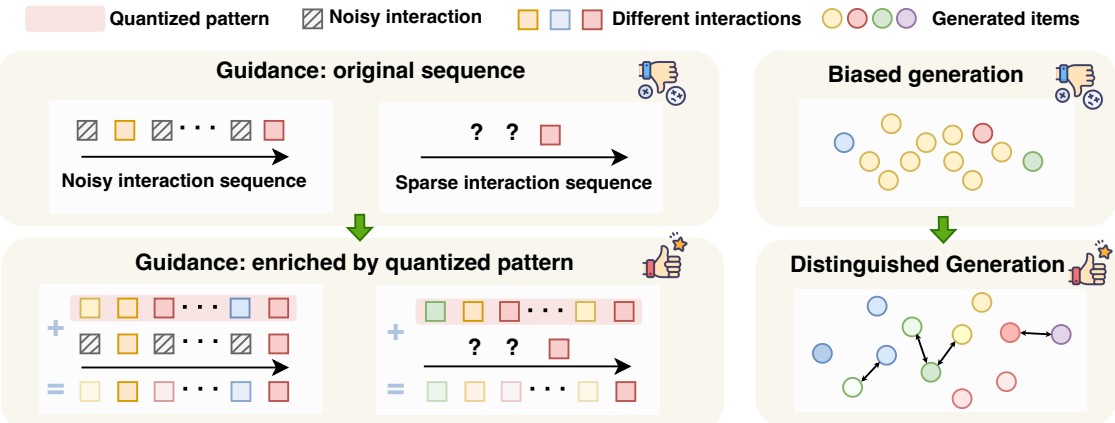

Figure 1: Challenges in adapting DMs to the sequential recommendation: (left) the heterogeneous (*e.g.*, sparse) or noisy (*e.g.*, mis-click) sequences as guidance, and (right) the biased generation in the item embedding space.

items and their patterns. In terms of recommendation performance, this would result in the generation of unbalanced items as shown on the right side of Figure 1, which may amplify the bias [25, 26, 43] and restrict DMs from meeting the personalized interests of different users. Ideally, different users have their own personalized interests [35], which requires the generation to explicitly distinguish these differences. Even in the case where two users interact with the same item, we believe it is reasonable to assume that they may reach this item from different perspectives.

To tackle the aforementioned problems, we propose Distinguished Quantized Guidance for Diffusion-based Sequence Recommendation (DiQDiff). Specifically, we introduce a Semantic Vector Quantization (SVQ) module to quantize sequences into semantic vectors (that encodes collaborative signals and category interests) with a discrete codebook. As demonstrated in Figure 1, by combining the quantized vectors with original sequences, the guidance can be enhanced with underlying semantic patterns. Intuitively, the enhanced guidance information can provide recognizable information even with sparse interaction sequences and provide a smoothed representation given noisy signals. On the other hand, to distinguish personalized views and mitigate biased generation, we design a Contrastive Discrepancy Maximization (CDM) module, which pushes away the denoised item representations from different user interaction sequences with contrastive loss. In practice, the quantization module (*i.e.*, SVQ) may introduce extra risks generating biased results since the codebook itself may reduce the utilization of more precise signals. Fortunately, the CDM can prevent DMs from generating similar items for different users, which forces the model to learn the different patterns from the enhanced guidance. We conducted experiments to validate the effectiveness of DiQDiff by comparing the recommendation performance with multiple baseline models, including both traditional recommenders and generative recommenders. Extensive experimental results demonstrate that DiQDiff achieves state-of-the-art among multiple leading approaches across four benchmark datasets.

We summarize the contribution of this paper as follows:

- We identify the challenges of heterogeneous and noisy guidance, and biased generation in diffusion-based recommender systems, and propose a novel framework DiQDiff to address them.
- To the best of our knowledge, DiQDiff is the first work to investigate the combination of guidance vector quantization and distinguished generation in DMs for sequential recommendation.
- We conducted extensive experiments in four public datasets, and the results demonstrate the superiority of our method.

## 2 Related Work

### 2.1 Sequential Recommendation

Sequential recommendation (SR) formulates a next-item prediction task which aims to capture user preferences based on the historical interaction sequence and predict his/her next interaction. Traditional SR solutions that have been widely adopted in practice are discriminative models such as GRU4Rec [9], Time-LSTM [47], and SASRec [15]. They represent items in representation space and learn to predict the next item based on interaction sequences while keeping the decision different from sampled negatives. Recently, researchers have found that the next-item recommendation can also be formulated as an item generation task, taking advantage of the superior distribution modeling ability of generative models such as VAE [30, 34, 42], GANs [6, 33], and Diffusion models [3, 18, 27, 44]. Among these techniques, DM-based recommenders [22, 24, 39, 44] have recently seen notable advances due to their ability to model complex distributions and generate high-quality samples. Specifically, DMs are used to model and generate the next-item representation by corrupting them with Gaussian noise and denoising them step by step guided by the historical sequence. In this work, we focus on the problems within DM-based sequential recommendation, which emphasize the importance of extracting robust guidance and personalized item generation.

### 2.2 Vector Quantization for Generative Models

In generative models, Vector Quantization (VQ) learns a codebook to discretize input representation (*e.g.*, images or audios) into code

vectors [1, 13, 37, 45] aiming to enhance the model's semantic extraction ability. During training, the main objective is to improve the reconstruction or generation accuracy of input from these compressed codebook representations. For instance, VQVAE [37] maps images into latent features and then quantizes them with the nearest code vectors in the codebook. Finally, the decoder reconstructs the original images based on the quantized representation. VQGAN [4] generates images with quantized representation from the learned codebook, while the discriminator distinguishes between real and generated images. VQDiffusion [7, 11] quantizes images based on the pre-trained VQVAE and then reconstructs images with discrete diffusion models. Unlike these methods which quantize the input images directly, in the sequential recommender task, we quantize the guidance that encodes the user's personal interests and reconstruct the items to recommend with DMs.

## 2.3 Vector Quantization for RSs

In recommender systems, vector quantization (VQ) techniques can identify shared patterns or category information across representations (*i.e.*, items or users). As one of the most representative solutions, [28] learns a codebook to identify user interest clusters, and uses this extra semantic information to enhance the click-through rate prediction performance. In basket recommendation, NPA [23] learns to encode the common item combination patterns into a codebook for effectively capturing and identifying users' shopping intentions. CAGE [21] further improves this idea to generate user and item category trees, simultaneously learning the item and user representations in an end-to-end manner. However, integrating vector quantization techniques into generative recommenders ( especially DM-based recommenders) remains largely unexplored, and we propose to quantize the guidance of DMs to understand user interests better and provide a more robust guidance representation against heterogeneous and noisy user histories.

## 2.4 Bias generation in DMs

Diffusion models have gained widespread attention for modeling complex data distribution. However, they often inherit and amplify the biases [25, 26, 31] present in the original training data during the generation process, which leads to a biased generation [12]. Thus promoting the diversity of generation [12, 20, 25, 31] has become an important direction in current research.

## 3 Preliminary

### 3.1 Task Formulation

Let $\mathcal{I}$ be the item set, $s = [x_1, x_2, \ldots, x_{L-1}]$ be the interaction sequence for a user, $x_L$ be the ground-truth next item that the user will interact with, where $x_l \in \mathcal{I}$ is the $l$-th interaction in the chronological sequence. The sequential recommendation aims to recommend the item that best aligns with user interests as the next item $x_L$ based on the historical interaction sequence $s$.

### 3.2 Denoising Diffusion Probabilistic Models

Denoising Diffusion Probabilistic Models (DDPM) [10] is a generative model designed with two Markov processes, consisting of a forward process that diffuses the input into random noise and a reverse process that recovers the input back from the random noise.

**Forward process** corrupts the input $\mathbf{x}^0$ by adding Gaussian noise step by step with a Markov Chain. Formally, the forward transition from $\mathbf{x}^{t-1}$ to $\mathbf{x}^t$ can be defined as a Gaussian noise injection function $q(\mathbf{x}^t|\mathbf{x}^{t-1}) = \mathcal{N}(\mathbf{x}^t; \sqrt{1-\beta_t}\mathbf{x}^{t-1}, \beta_t\mathbf{I})$, where $t \in \{1, \ldots, T\}$ denotes the diffusion step, and $[\beta_1, \beta_2, \ldots, \beta_T]$ denote the variance schedule. Let $\alpha_t = 1 - \beta_t, \bar{\alpha}_t = \prod_{t'=1}^{t} \alpha_{t'}$, we can derive $\mathbf{x}^t = \sqrt{\bar{\alpha}_t}\mathbf{x}^0 + \sqrt{1-\bar{\alpha}_t}\boldsymbol{\epsilon}$, where $\boldsymbol{\epsilon} \sim \mathcal{N}(\mathbf{0}, \mathbf{I})$ [10]. At the final step $T$, the $x^T$ approximates a pure Gaussian noise.

**Reverse process** eliminates the noise step by step to recover $\mathbf{x}^0$ from $\mathbf{x}^T \sim \mathcal{N}(\mathbf{0}, \mathbf{I})$ with another Markov Chain. Formally, the denoising transition from $\mathbf{x}^t$ to $\mathbf{x}^{t-1}$ can be defined as $p_\theta(\mathbf{x}^{t-1}|\mathbf{x}^t) = N(\mathbf{x}^{t-1}; \boldsymbol{\mu}_\theta(\mathbf{x}^t, t), \Sigma_\theta(\mathbf{x}^t, t))$, where $\boldsymbol{\mu}_\theta(\mathbf{x}^t, t)$ and $\Sigma_\theta(\mathbf{x}^t, t)$ are the predicted mean and covariance from neural network parameterized by $\theta$. When $p_\theta$ successfully approximates the real distribution after training, DDPM can generate $\mathbf{x}^0$ step by step from the initial Gaussian noise during inference. According to [10], the **optimization objective** for $\theta$ is the variational bound of negative log-likelihood $-\log p_\theta(\mathbf{x}^0)$, which is the KL divergence between $q(\mathbf{x}^{t-1} \mid \mathbf{x}^t, \mathbf{x}^0)$ and $p_\theta(\mathbf{x}^{t-1}|\mathbf{x}^t)$:

$$\mathcal{L} = \underbrace{D_{KL}\left(q\left(\mathbf{x}^T \mid \mathbf{x}^0\right) \| p\left(\mathbf{x}^T\right)\right)}_{\mathcal{L}_T} - \underbrace{\mathbb{E}_{q(\mathbf{x}^1|\mathbf{x}^0)}\left[\log_\theta\left(\mathbf{x}^0 \mid \mathbf{x}^1\right)\right]}_{\mathcal{L}_0}$$
$$+ \sum_{t=2}^{T} \underbrace{E_{q(\mathbf{x}^t|\mathbf{x}^0)}\left[D_{KL}\left(q\left(\mathbf{x}^{t-1} \mid \mathbf{x}^t, \mathbf{x}^0\right) \| p_\theta\left(\mathbf{x}^{t-1} \mid \mathbf{x}^t\right)\right]}_{\mathcal{L}_{t-1}}, \quad (1)$$

where $q\left(\mathbf{x}^{t-1} \mid \mathbf{x}^t, \mathbf{x}^0\right) = \mathcal{N}\left(\mathbf{x}^{t-1}; \tilde{\boldsymbol{\mu}}_t\left(\mathbf{x}^t, \mathbf{x}^0\right), \tilde{\beta}_t\mathbf{I}\right)$ is the posterior distribution, and we have:

$$\tilde{\boldsymbol{\mu}}_t\left(\mathbf{x}^t, \mathbf{x}^0\right) = \frac{\sqrt{\bar{\alpha}_{t-1}}\beta_t}{1-\bar{\alpha}_t}\mathbf{x}^0 + \frac{\sqrt{\alpha_t}\left(1-\bar{\alpha}_{t-1}\right)}{1-\bar{\alpha}_t}\mathbf{x}^t, \quad (2)$$

$$\tilde{\beta}_t = \frac{1-\bar{\alpha}_{t-1}}{1-\bar{\alpha}_t}\beta_t. \quad (3)$$

According to the parameterization in [10], we have $\boldsymbol{\mu}_\theta(\mathbf{x}^t, t) = \frac{1}{\sqrt{\alpha_t}}\left(\mathbf{x}^t - \frac{1-\alpha_t}{\sqrt{1-\bar{\alpha}_t}}\boldsymbol{\epsilon}_\theta(\mathbf{x}^t, t)\right)$, and the loss of Equation 1 can be further simplified as below:

$$\mathcal{L}_{\text{simple}}(\theta) := \mathbb{E}_{t,\mathbf{x}_0,\boldsymbol{\epsilon}}\left[\left\|\boldsymbol{\epsilon} - \boldsymbol{\epsilon}_\theta(\sqrt{\bar{\alpha}_t}\mathbf{x}^0 + \sqrt{1-\bar{\alpha}_t}\boldsymbol{\epsilon}, t)\right\|^2\right], \quad (4)$$

where $\boldsymbol{\epsilon} \sim \mathcal{N}(\mathbf{0}, \mathbf{I})$, $t$ is uniform between 1 and $T$, and $\boldsymbol{\epsilon}_\theta\left(\mathbf{x}^t, t\right)$ is the predicted noise added in the forward process with neural network (*e.g.*, U-Net [40] or Transformer [29]). Intuitively, this transforms the problem into denoising score matching across noise at $t$ steps.

## 4 Method

### 4.1 Overview of DiQDiff

We follow existing diffusion-based SR approaches [18, 44] which consists of a personalized guidance extraction and a diffusion-based item generation phase. As demonstrated in Figure 2, our proposed DiQDiff introduces two key components: Semantic Vector Quantization (SVQ) and Contrastive Discrepancy Maximization (CDM). The SVQ module uses a codebook quantization strategy to provide

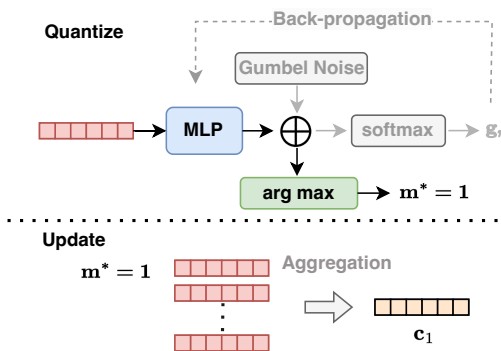

Figure 2: The framework of DiQDiff. The Semantic Vector Quantization is applied to quantize sequences with a semantic codebook, extracting accurate and robust guidance. The Contrastive Discrepancy Maximization is utilized to maximize the distance between different denoising trajectories, enabling distinguished item generation for different users.

accurate and robust guidance for DMs, consequently addressing the challenge of heterogeneous and noisy guidance; The CDM module distinguishes denoised items from different sequences with contrastive loss to handle the biased generation challenge. During training, DiQDiff introduces Gaussian noise to the ground-truth next items in the forward process. Then, we enhance the guidance by extracting quantized embeddings from SVQ. Subsequently, the denoising model is trained to recover the corrupted items conditioned on the enhanced guidance, optimizing both the reconstruction loss from DMs and the contrastive loss from the CDM. During inference, pure Gaussian noise serves as the input, allowing the trained denoising model to generate the next items step by step based on the guidance from SVQ. We summarize the training and inference processes of DiQDiff in Algorithm 1 and 2 respectively, and detail the related technologies in the following sections.

## 4.2 Guidance Extraction with SVQ

As illustrated in section 1, user behavior sequences can be sparse or noisy. Merely using the original interaction sequence as guidance makes it challenging for DMs to understand user interest. To extract accurate and robust guidance for DMs, we adopt SVQ to extract semantic features (*e.g.*, category interests) from collaborative records and maintain a corresponding codebook for the semantic vectors.

Specifically, we first transform each item $v \in \mathcal{I}$ into its corresponding embedding $\mathbf{x} \in \mathbb{R}^D$, where $D$ denotes the embedding dimension. Consequently, the sequence $s = [x_1, x_2, \ldots, x_{L-1}]$ can be represented as $\mathbf{s} = [\mathbf{x}_1, \mathbf{x}_2, \ldots, \mathbf{x}_{L-1}] \in \mathbb{R}^{(L-1) \times D}$, and the next item is represented as $\mathbf{x}_L$. We then define the semantic codebook as $\mathbf{C} = \{\mathbf{c}_m\}_{m=1}^{M}$, where each code vector $\mathbf{c}_m \in \mathbb{R}^{(L-1) \times D}$ matches the size of the sequence embedding, and $M$ is the number of discrete code vectors in the codebook.

Given a codebook, one may follow a deterministic quantization strategy that simply selects the nearest code vector with "arg min" [37], but this would introduce a nondifferentiable step. Instead, we employ stochastic quantization [13, 45] to sample from a predicted

vector distribution, which enables end-to-end training. As shown in Figure 3, we implement a code selection model $f_\varphi(\cdot)$ with an MLP to compute the $M$-dimensional logits for each sequence $\mathbf{s}$. Then we utilize the Gumbel-Softmax technique [1, 14, 45] to select the discrete code vector for the sequence, facilitating back-propagation. Formally, we have:

$$\mathbf{o} = f_\varphi(\mathbf{s}), \mathbf{o} \in \mathbb{R}^M, \tag{5}$$

$$g_m = \frac{\exp((o_m + n_m)/\tau)}{\sum_{m'=1}^{M} \exp((o_{m'} + n_{m'})/\tau)}, \tag{6}$$

where $\tau$ is the temperature, $n_m \sim \text{Gumbel}(0, 1)$, whose density function is $e^{-(n+e^{-n})}$, and $g_m \in [0, 1]$. In the forward propagation of training, we adopt $m^* = \arg\max_m g_m$ to select the $m^*$-th code vector for quantizing the sequence $\mathbf{s}$, and we have $\mathbf{s}_q = \mathbf{c}_{m^*}$. During training, we utilize the gradient from the Gumbel-Softmax to further backpropagate towards the code selection model $f_\varphi(\cdot)$.

After obtaining the quantized code $\mathbf{s}_q$ for the sequence $\mathbf{s}$, we then combine it with the original sequence:

$$\tilde{\mathbf{s}} = \lambda_q \mathbf{s}_q + \mathbf{s}, \tag{7}$$

Figure 3: Quantization and updating process in SVQ.

where $\lambda_q \in [0, 1]$ controls the injection strength of the quantized vector $\mathbf{s}_q$, and the combined representation $\tilde{\mathbf{s}}$ will serve as the enhanced guidance for DMs. Intuitively, for sparse sequences with insufficient interactions, the closest code would provide extra information that best aligns with the user's interest; and for noisy sequences, the extracted code would help amplify recognizable patterns and reduce the influence of irrelevant noises, which improves the expressiveness of the guidance.

In addition to the quantization in SVQ, we update the semantic codebook with expectation-maximization which is widely used in clustering methods. As illustrated in Figure 3, we aggregate the sequences that extract the same code vector, and use the aggregated result to update the corresponding code vector:

$$\mathbf{c}'_m = \frac{1}{|S_m|} \sum_{\mathbf{s} \in S_m} \mathbf{s}, \tag{8}$$

where $S_m$ denotes the set of sequences in the batch samples that select $m$-th code. This means that the code vectors maintain the most representative information about the semantic cluster (*e.g.*, collaborative signals and category interests).

## 4.3 Distinguished Generation with CDM

After extracting the guidance $\tilde{\mathbf{s}}$ as detailed in Section 4.2, DiQDiff adopts a conditional DDPM to train the denoising model, then denoise step-by-step to generate the next items conditioned on the guidance during inference. To enable the distinguished generation of items for personalized interests, we introduce the CDM module to push away denoised items from different guidance sequences with contrastive loss.

Specifically, we first add Gaussian noise to the ground truth next item in the forward process:

$$\mathbf{x}_L^t = \sqrt{\bar{\alpha}_t}\mathbf{x}_L + \sqrt{1 - \bar{\alpha}_t}\boldsymbol{\epsilon}, \quad t \in \{1, \dots, T\}. \tag{9}$$

Following [18, 39, 44], rather than predicting the noise added in the forward process, we estimate the target item $\hat{\mathbf{x}}_L^0$ under the guidance $\tilde{\mathbf{s}}$ at each time step:

$$\hat{\mathbf{x}}_L^0 = f_\theta(\mathbf{x}_L^t, \tilde{\mathbf{s}}, t), \tag{10}$$

where the $f_\theta(\cdot)$ is implemented by a Transformer following the prior study [18]. Then, the loss in Equation 4 can be reformulated:

$$\mathcal{L}_r = \mathbb{E}_{t, \mathbf{x}_0, \boldsymbol{\epsilon}} \left[ \left\| \mathbf{x}_L - f_\theta(\sqrt{\bar{\alpha}_t}\mathbf{x}_L + \sqrt{1 - \bar{\alpha}_t}\boldsymbol{\epsilon}, \tilde{\mathbf{s}}, t) \right\|^2 \right], \tag{11}$$

where $\mathbf{x}_L$ is the ground-truth, $\mathcal{L}_r$ denotes the reconstruction loss.

To prevent DMs from biased item generation, we propose to maximize the difference between the predicted item representation $\hat{\mathbf{x}}_L^0$ from different sequences with contrastive loss. Formally, given denoised item representations $\hat{\mathbf{x}}_L^0$ and $\hat{\mathbf{x}}_L'^0$ from different sequences in the batch $B_x$, the CDM loss can be defined as below:

$$\mathcal{L}_c = \mathbb{E}_{\hat{\mathbf{x}}_L^0} \left[ \log \sum_{\hat{\mathbf{x}}_L'^0 \in B_x} \left[ \exp \left( \mathrm{sim}(\hat{\mathbf{x}}_L^0, \hat{\mathbf{x}}_L'^0) \right) \right] \right], \tag{12}$$

where $\mathrm{sim}(\cdot)$ denotes the cosine similarity function. Minimizing $\mathcal{L}_c$ will push away the denoised items from different sequences, thus realizing distinguished generations for different users' personalized

interests. Finally, combining it into the total loss for training the denoising model $f_\theta(\cdot)$, we have:

$$\mathcal{L} = \mathcal{L}_r + \lambda_c \mathcal{L}_c, \tag{13}$$

where $\lambda_c$ denotes the strength coefficient of CDM in the optimizing objective. Note that the training will simultaneously optimize the denoising model $f_\theta(\cdot)$ and the code selection model $f_\varphi(\cdot)$ in the end-to-end design.

During inference, we can generate items $\mathbf{x}_L^0$ by denoising the Gaussian noise step-by-step. According to Equation 2, we have the transformed stepwise output as:

$$\mathbf{x}_L^{t-1} = \frac{\sqrt{\bar{\alpha}_{t-1}}\beta_t}{1 - \bar{\alpha}_t} f_\theta(\mathbf{x}_L^t, \tilde{\mathbf{s}}, t) + \frac{\sqrt{\alpha_t}(1 - \bar{\alpha}_{t-1})}{1 - \bar{\alpha}_t}\mathbf{x}_L^t + \sqrt{\tilde{\beta}_t}\mathbf{z}, \tag{14}$$

where $\mathbf{x}_L^T$ is a pure Gaussian noise, $\mathbf{z} \sim \mathcal{N}(\mathbf{0}, \mathbf{I})$. And note that $f_\theta(\cdot)$ knows how to generate different denoising trajectories for $\mathbf{x}_L^T$ and $\mathbf{x}_L^0$ given that they come from different users. Finally, with the generated item representation $\mathbf{x}_L^0$, we calculate the inner product between this representation and all item embeddings in the candidate set, then top-K nearest items are selected as recommendation.

---

**Algorithm 1:** Training process of DiQDiff

---

**Input:** Sequence $\mathbf{s}$, next item $\mathbf{x}_L$, codebook $\mathbf{C}$, hyperparameters $\lambda_q, \lambda_c$, variance schedule $[\alpha_t]_{t=1}^T$

**Output:** Optimal denoising model $f_\theta(\cdot)$ and optimal code selection model $f_\varphi(\cdot)$.

1: **repeat**
2:     $t \sim \{1, \dots, T\}$, $\boldsymbol{\epsilon} \sim \mathcal{N}(0, I)$     ▷ Sample diffusion step and Gaussian noise.
3:     $\mathbf{x}_L^t = \sqrt{\bar{\alpha}_t}\mathbf{x}_L + \sqrt{1 - \bar{\alpha}_t}\boldsymbol{\epsilon}$     ▷ Add Gaussian noise.
4:     $\mathbf{s}_q \leftarrow$ quantize $\mathbf{s}$ with SVQ.
5:     $\mathbf{C} \leftarrow$ Equation 8     ▷ Update the codebook.
6:     $\tilde{\mathbf{s}} = \mathbf{s} + \lambda_q \mathbf{s}_q$     ▷ Enhance the guidance with $\mathbf{s_q}$.
7:     $\mathcal{L}_r, \mathcal{L}_c \leftarrow$ Equantion 11 and 12.
8:     $\mathcal{L} = \mathcal{L}_r + \lambda_c \mathcal{L}_c$.
9:     $\theta = \theta - \mu \nabla_\theta \mathcal{L}$,   $\varphi = \varphi - \mu \nabla_\varphi \mathcal{L}$.
10: **until** converged

---

**Algorithm 2:** Inference process of DiQDiff

---

**Input:** Sequence $\mathbf{s}$, hyperparameters $\lambda_q$, codebook $\mathbf{C}$, optimal denoising model $f_\theta(\cdot)$, and code selection model $f_\varphi(\cdot)$.

**Output:** Generated item $\mathbf{x}_L^0$.

1:   $\mathbf{x}_L^T \sim \mathcal{N}(0, I)$     ▷ Sample Gaussian noise.
2:   $\mathbf{s}_q \leftarrow$ quantize $\mathbf{s}$ with SVQ.
3:   $\mathbf{C} \leftarrow$ Equation 8     ▷ Update the codebook.
4:   $\tilde{\mathbf{s}} = \mathbf{s} + \lambda_q \mathbf{s}_q$     ▷ Enhance the guidance with $\mathbf{s_q}$.
5:   **for** $t = T, \dots, 1$ **do**
6:     $\hat{\mathbf{x}}_L^0 = f_\theta(\mathbf{x}_L^t, \tilde{\mathbf{s}}, t)$.
7:     $\mathbf{x}_L^{t-1} = \frac{\sqrt{\bar{\alpha}_{t-1}}\beta_t}{1 - \bar{\alpha}_t}\hat{\mathbf{x}}_L^0 + \frac{\sqrt{\alpha_t}(1 - \bar{\alpha}_{t-1})}{1 - \bar{\alpha}_t}\mathbf{x}_L^t + \sqrt{\tilde{\beta}_t}\mathbf{z}$ .
8:   **end for**
9:   **return** $\mathbf{x}_L^0$

---

**Table 1: Overall performance of different methods for the sequential recommendation. The highest score in each row is typed in bold to indicate statistically significant improvements (p < 0.05), while the second-best score is underlined. We use "H" and "N" to represent HR and NDCG respectively.**

| Dataset | Metric | GRU4Rec | SASRec | BERT4Rec | ComiRec | TiMiRec | STOSA | DuoRec | CL4SRec | ACVAE | DreamRec | DiffuRec | DiQDiff |
|---------|--------|---------|--------|----------|---------|---------|-------|--------|---------|-------|----------|----------|---------|
| ML-1M | H@5 | 5.11 | 9.38 | 13.64 | 6.11 | 16.21 | 7.05 | 13.7 | 12.61 | 12.72 | 16.05 | 16.02 | **16.44** |
| | H@10 | 10.17 | 16.89 | 20.57 | 12.04 | 23.71 | 14.39 | 21.41 | 20.17 | 19.93 | 24.63 | 24.28 | **24.92** |
| | H@20 | 18.70 | 28.32 | 29.95 | 21.01 | 33.23 | 24.99 | 32.97 | 31.91 | 28.97 | 35.84 | 35.63 | **36.10** |
| | N@5 | 3.05 | 5.32 | 8.89 | 3.52 | 10.88 | 3.72 | 7.92 | 7.58 | 8.23 | 10.61 | 10.41 | **10.90** |
| | N@10 | 4.68 | 7.73 | 11.13 | 5.41 | 13.31 | 6.08 | 10.59 | 10.02 | 10.54 | 13.36 | 13.05 | **13.61** |
| | N@20 | 6.82 | 10.59 | 13.48 | 7.65 | 15.70 | 8.72 | 13.50 | 12.97 | 12.82 | 16.20 | 15.90 | **16.43** |
| Beauty | H@5 | 1.01 | 3.27 | 2.13 | 2.05 | 1.90 | 3.55 | 5.37 | 5.25 | 2.47 | 5.31 | 5.33 | **5.64** |
| | H@10 | 1.94 | 6.26 | 3.72 | 4.45 | 3.34 | 6.20 | 7.63 | 7.29 | 3.88 | 7.13 | 7.21 | **7.82** |
| | H@20 | 3.85 | 8.98 | 5.79 | 7.70 | 5.17 | 9.59 | 10.72 | 10.58 | 6.12 | 10.62 | 10.51 | **10.93** |
| | N@5 | 0.61 | 2.40 | 1.32 | 1.05 | 1.24 | 2.56 | 3.28 | 3.03 | 1.69 | 3.28 | 3.82 | **4.04** |
| | N@10 | 0.90 | 3.23 | 1.83 | 1.83 | 1.70 | 3.21 | 4.19 | 3.99 | 2.14 | 4.19 | 4.43 | **4.74** |
| | N@20 | 1.38 | 3.66 | 2.35 | 2.65 | 2.16 | 3.76 | 4.99 | 4.85 | 2.70 | 4.99 | 5.34 | **5.52** |
| Toys | H@5 | 1.10 | 4.53 | 1.93 | 2.30 | 1.16 | 4.22 | 5.66 | 5.48 | 2.19 | 5.47 | 5.58 | **6.04** |
| | H@10 | 1.85 | 6.55 | 2.93 | 4.29 | 1.82 | 6.94 | 7.16 | 6.87 | 3.07 | 7.25 | 7.22 | **7.70** |
| | H@20 | 3.18 | 9.23 | 4.59 | 6.94 | 2.72 | 9.51 | 9.81 | 10.09 | 4.41 | 9.68 | 9.84 | **10.28** |
| | N@5 | 0.70 | 3.01 | 1.16 | 1.16 | 0.71 | 3.10 | 3.11 | 3.34 | 1.56 | 4.04 | 4.18 | **4.47** |
| | N@10 | 0.94 | 3.75 | 1.49 | 1.80 | 0.91 | 3.88 | 3.92 | 4.27 | 1.85 | 4.62 | 4.75 | **5.00** |
| | N@20 | 1.27 | 4.33 | 1.90 | 2.46 | 1.14 | 4.38 | 4.71 | 5.08 | 2.18 | 5.22 | 5.35 | **5.65** |
| Steam | H@5 | 3.01 | 4.74 | 4.74 | 2.29 | 6.02 | 4.85 | 5.69 | 5.62 | 5.58 | 5.96 | 6.72 | **7.13** |
| | H@10 | 5.43 | 8.38 | 7.94 | 5.44 | 9.67 | 8.59 | 9.78 | 9.45 | 9.28 | 9.68 | 10.51 | **11.41** |
| | H@20 | 9.23 | 13.61 | 12.73 | 10.37 | 14.89 | 14.11 | 15.61 | 15.06 | 14.48 | 15.08 | 16.09 | **17.57** |
| | N@5 | 1.83 | 2.88 | 2.97 | 1.10 | 3.87 | 2.92 | 3.36 | 3.48 | 3.54 | 3.84 | 4.19 | **4.61** |
| | N@10 | 2.60 | 4.05 | 4.00 | 2.11 | 5.04 | 4.12 | 4.68 | 4.71 | 4.73 | 5.03 | 5.50 | **5.98** |
| | N@20 | 3.56 | 5.36 | 5.20 | 3.34 | 6.36 | 5.51 | 6.14 | 6.12 | 6.04 | 6.39 | 7.11 | **7.50** |

## 5 Experiments

In this section, we conduct extensive experiments to validate the effectiveness of DiQDiff, answering the following questions:

- RQ1: How does DiQDiff perform compared with multiple baseline models in the sequential recommendation?
- RQ2: How does the design of SVQ and CDM bring improvements to DiQDff, respectively?
- RQ3: How sensitive is DiQDiff to different settings (*i.e.*, codebook size, strength of the SVM, and that of CDM )?

### 5.1 Experimental Settings

*5.1.1 **Datasets**.* We conduct experiments across four widely-used datasets in sequential recommendation. ML-1M is a movie dataset that includes one million ratings from 6,000 users across 4,000 films. The Amazon Beauty and Amazon Toys datasets consist of user reviews for beauty products and toys collected from the Amazon platform over nearly 20 years. The Steam dataset gathers information about video games available on the Steam platform, encompassing users' playing time, prices, categories, and more. Following previous studies [15, 18, 36], the user-item interactions are organized chronologically based on the timestamps, and those with fewer than five interactions are filtered out. The statistics of these datasets are listed in Table 2, exhibiting notable differences in sequence lengths and dataset sizes in real-world scenarios.

*5.1.2 **Baseline**.* We compare DiQDiff with a variety of leading approaches in sequential recommendation, including traditional

**Table 2: Stastics of the four datasets.**

| Dataset | Sequence | items | Avg-len |
|---------|----------|-------|---------|
| ML-1M | 6,040 | 3,416 | 165.50 |
| Beauty | 22,363 | 12,101 | 8.53 |
| Toys | 19,412 | 11,924 | 8.63 |
| Steam | 281,428 | 13,044 | 12.40 |

recommenders, interest learning methods, contrastive-based methods, and generative recommenders.

- **Traditional recommenders:** GRU4Rec [9], SASRec [15], and Bert4Rec [36] predict the next-item with discriminative models such as GRU [9] and Transformer [15], which can capture the preference dependency in sequences.
- **Interest learning methods:** ComiRec [2] and TiMiRec [38] aim to capture users' multiple interests through modules like dynamic routing. STOSA [5] focuses on users' dynamic interests by employing stochastic embeddings.
- **Contrastive-based methods:** DuoRec [32] and CL4SRec [41] propose different augmentation techniques and adopt contrastive learning to alleviate the representation degeneration or data sparsity problem in SR;
- **Generative recommenders:** ACVAE [42] introduces an Adversarial and Contrastive Variational Autoencoder to generate high-quality latent representations for SR. DreamRec [44] and

**Table 3: Results of ablation experiments. The best results are highlighted in bold, while the second-best are underlined. "Base" refers to the DiQDiff variant without both SVQ and CDM, while "w/o SVQ" and "w/o CDM" indicate the variants of DiQDiff that exclude SVQ or CDM, respectively. We use "H" and "N" to represent HR and NDCG respectively.**

| Dataset | Ablation | H@5 | H@10 | H@20 | N@5 | N@10 | N@20 |
|---------|----------|-----|------|------|-----|------|------|
| Beauty | Base | 5.41 | 7.62 | 10.66 | 3.90 | 4.61 | 5.38 |
| | w/o SVQ | 5.46 ↑ | 7.62 ↑ | 10.65 ↑ | 3.99 ↑ | 4.67 ↑ | 5.44 ↑ |
| | w/o CDM | 5.46 ↑ | 7.69 ↑ | 10.77 ↑ | 3.95 ↑ | 4.67 ↑ | 5.44 ↑ |
| | DiQDiff | **5.64** ↑ | **7.82** ↑ | **10.93** ↑ | **4.04** ↑ | **4.74** ↑ | **5.52** ↑ |
| Toys | Base | 5.65 | 7.41 | 9.85 | 4.17 | 4.74 | 5.35 |
| | w/o SVQ | 5.81 ↑ | 7.60 ↑ | 10.17 ↑ | 4.36 ↑ | 4.94 ↑ | 5.59 ↑ |
| | w/o CDM | 5.79 ↑ | 7.61 ↑ | 10.03 ↑ | 4.34 ↑ | 4.92 ↑ | 5.53 ↑ |
| | DiQDiff | **6.04** ↑ | **7.72** ↑ | **10.28** ↑ | **4.47** ↑ | **5.00** ↑ | **5.65** ↑ |
| ML-1M | Base | 15.45 | 24.15 | 35.68 | 10.21 | 13.00 | 15.90 |
| | w/o SVQ | 15.62 ↑ | 23.80 ↑ | 34.81 ↑ | 10.52 ↑ | 13.15 ↑ | 15.95 ↑ |
| | w/o CDM | 16.43 ↑ | 24.70 ↑ | **36.12** ↑ | **10.91** ↑ | 13.53 ↑ | 16.40 ↑ |
| | DiQDiff | **16.44** ↑ | **24.92** ↑ | 36.10 ↑ | 10.90 ↑ | **13.61** ↑ | **16.43** ↑ |
| Steam | Base | 6.70 | 10.90 | 16.75 | 4.30 | 5.65 | 7.12 |
| | w/o SVQ | 6.70 ↑ | 10.91 ↑ | 16.79 ↑ | 4.33 ↑ | 5.67 ↑ | 7.19 ↑ |
| | w/o CDM | 6.99 ↑ | 11.29 ↑ | 17.43 ↑ | 4.53 ↑ | 5.91 ↑ | 7.46 ↑ |
| | DiQDiff | **7.13** ↑ | **11.41** ↑ | **17.57** ↑ | **4.61** ↑ | **5.98** ↑ | **7.53** ↑ |

DiffuRec [18] utilize Denoising Diffusion Probabilistic Models (DDPM) to model item distribution, generating the next item through a denoising process guided by interaction sequences.

5.1.3 **Implementation Details**. Following the setting of previous works [15, 18], we employ the Adam optimizer, where the initial learning rate is 0.001. The embedding dimension is set to 128, and the batch size is 512. The dropout rates for the denoising model and item embeddings are set to 0.1 and 0.3 respectively. The number of time steps $T$ of DDPM is 32, and we utilize a truncated linear schedule for the noise schedule. Each method is evaluated over five trials, and the averaged results are reported. The maximum sequence length of ML-1M is set to 200 and that of the other three datasets is set to 50. Sequences with fewer interactions than the maximum length are padded with a padding token. The strengths $\lambda_q, \lambda_c$ of the SVM and CDM are varied within the range {0.2, 0.4, 0.6, 0.8, 1.0}, while the codebook size $M$ is selected from {4, 8, 16, 32, 64}. To evaluate the recommendation performance, we evaluate all models using Hit Rate (H@K) and Normalized Discounted Cumulative Gain (N@K), where $K = \{5, 10, 20\}$. Additionally, to ensure a fair comparison and efficient implementation, we evaluate diffusion-based recommenders (*i.e.,* DreamRec, DiffuRec, and DiQDiff) every two epochs and employ early stopping if the highest results remain unchanged over 10 evaluations.

## 5.2 Overall Performance (RQ1)

To answer Q1, we conducted experiments in all four datasets to compare the recommendation performance between DiQDiff and multiple baselines. We conducted each experiment for five times with different random seeds, and the averaged results are reported in Table 1. In general, diffusion-based recommenders (*i.e.,* DreamRec [44], DiffuRec [18], and DiQDiff) perform better than traditional recommenders (*e.g.,* GRU4Rec and SASRec) almost in all datasets and metrics, highlighting the effectiveness of DMs in modeling

item distributions and generating recommendations for the next step. Notably, DiQDiff consistently outperforms all benchmarks, achieving the highest Hit Rate and Normalized Discounted Cumulative Gain across four datasets. Especially on the largest steam dataset, DiQDiff significantly improves HR@20 and NDCG@20 by 9.2% and 5.5% respectively, compared to the best-performing baseline DiffuRec. The superiority of DiQDiff demonstrates the substantial effectiveness of our quantized guidance and distinguished generation in DMs for sequential recommendation.

## 5.3 Ablation Study (RQ2)

To answer Q2, we conduct an ablation study to validate the importance of SVQ and CDM respectively. The experimental results are presented in Table 3, where "Base" refers to the variant of DiQDiff without either SVQ or CDM, while "w/o SVQ" and "w/o CDM" represent the variants of DiQDiff that exclude SVQ or CDM, respectively. We observe that variants "w/o SVQ" and "w/o CDM" consistently outperform "Base" in all four datasets, indicating the feasibility of each individual design. Furthermore, DiQDiff demonstrates the highest performance among the three variants in most cases except a miner decrease of H@20 and N@5 in ML-1M. This suggests that the combination of the two components further improves the overall performance, indicating a superposable configuration.

To further illustrate the effectiveness of the components in the representation space, we plot the T-SNE of generated items' embeddings from different samples in Figure 4. We can see that the items generated from "Base" are unbalanced, indicating that DMs may inherit and amplify the biases presented in the data. In comparison, the items generated by variant "w/o SVQ' present the most balanced distribution, which means that CDM can effectively distinguish different item patterns during generation. Note that the "w/o CDM' variant only uses the SVQ component which does not necessarily mitigate the biased generation, it may potentially amplify the cluster-wise bias. In comparison, the combined solution

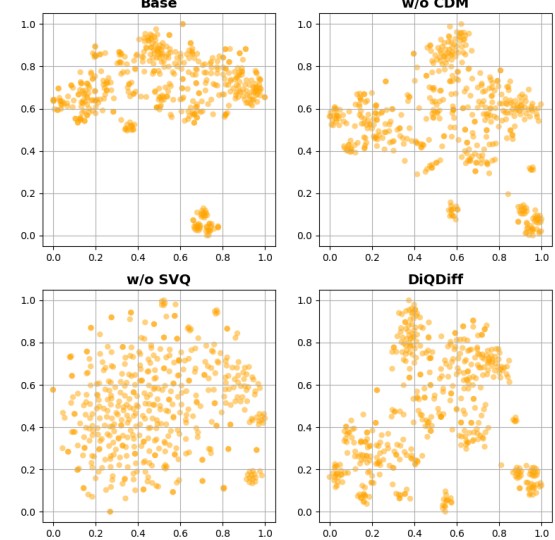

Figure 4: The T-SNE visualization of the generated item embeddings on the Toys dataset.

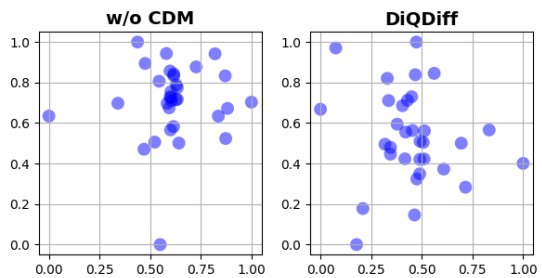

Figure 5: The T-SNE visualization displays the discrete code vectors in a codebook with $M = 32$ on the Toys dataset.

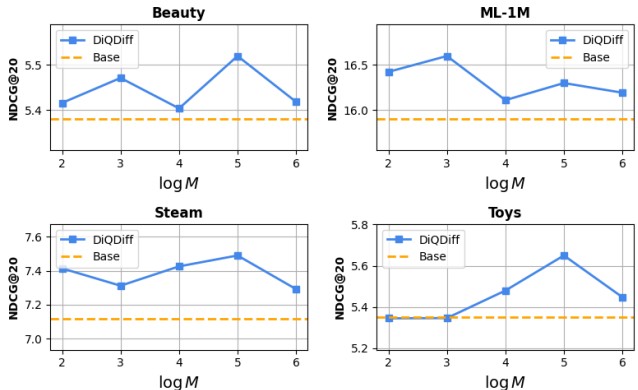

Figure 6: The sensitivity of DiQDiff to the hyperparameter $M$, which represents the codebook size.

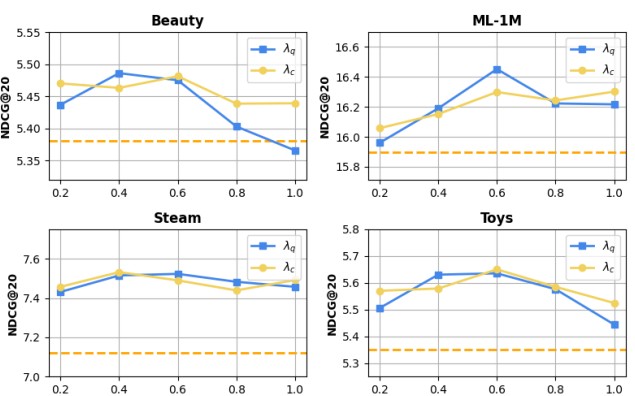

Figure 7: The sensitivity of DiQDiff to the hyperparameter $\lambda_q$ and $\lambda_c$.

DiQDiff still exhibits a certain degree of clustered structure in the distribution, but items are more distinguished with the existence of CDM. To further investigate the interaction between the two components, we visualize the codebook embedding learned by the variant "w/o CDM" and our DiQDiff, as shown in Figure 5. The code vectors from DiQDiff are more distinguished than those from variant "w/o CDM", validating that CDM can not only distinguish item representations, but can also back-propagate this discrepancy to the enhanced guidance and the semantic patterns in the codebook. We believe that this characteristic potentially helps in learning a more comprehensive and expressive codebook.

## 5.4 Sensitivity Analysis (RQ3)

To answer Q3, we further evaluate the sensitivity of DiQDiff hyperparameters $M$ (*i.e.,* the codebook size), $\lambda_q$ (*i.e.,* the injection strength of quantized vectors from SVQ in the guidance), and $\lambda_c$ (*i.e.,* the strength coefficient of CDM in the optimizing objective). As shown in Figure 6, the recommendation performance NDCG@20 of DiQDiff outperforms the variant "Base" stably, but the best point of $M$ varies across different datasets. We then present the curves of $\lambda_q$

and $\lambda_c$ analysis in Figure 7. Intuitively, increasing $\lambda_q$ would introduce semantically profound information to the sequence encoding, but over-injection may also dominate the guidance and suppress the information in the original user sequence. This is reflected in the increase-and-drop curve in Figure 7 across all datasets. Additionally, we also observe a similar pattern for $\lambda_c$, which indicates a potential optimal balancing point between a more distinguished item generation strategy and a more data-aligned strategy.

## 5.5 Conclusion

In this paper, we identify the challenge of heterogeneous and noisy guidance, as well as the biased generation challenge in diffusion-based recommender systems. To mitigate the problem, we propose a novel framework DiQDiff that first introduces a semantic vector quantization (SVQ) to enhance sparse and noisy sequences, then includes contrastive discrepancy maximization (CDM) to distinguish item generation and codebook representations. While we have provided evidence of DiQDiff's effectiveness in sequential recommendation tasks, the combination of SVQ and CDM may potentially benefit other tasks that encounter similar challenges.

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
