# OpenReview forum: "Distinguished Quantized Guidance for Diffusion-based Sequence Recommendation"
_ACM.org/TheWebConf/2025/Conference — WWW 2025 Oral_

### Official Review · Reviewer_J9tC · 2024-11-14

**Novelty:** 5
**Technical Quality:** 5

**Review:**

Summary:

This work identifies two key issues in diffusion-based recommender systems. First, the sequences are often heterogeneous in length and content (Heterogeneous and Noisy Guidance), and second, the diffusion models (DMs) are prone to data bias and tend to generate only the popular items (Biased Generation). To address these 2 issues, they propose Distinguished Quantized Guidance for Diffusion-based Sequence Recommendation (DiQDiff), which aims to extract robust guidance to understand user interests and generate distinguished items for personalized user interests within DMs. This framework first introduces a semantic vector quantization (SVQ) to enhance sparse and noisy sequences, then includes contrastive discrepancy maximization (CDM) to distinguish item generation and codebook representations.



Technical Quality:

The paper shows strong technical knowledge, using advanced ideas like quantized embeddings, contrastive discrepancy maximization, and sequence denoising, which reflect the authors' expertise. The preliminary section is well-organized, covering the task setup, the denoising model, and how the loss function is designed, making it easier to understand the method. The DiQDiff framework is structured clearly, and the diagrams help explain the concepts. The experiments are thorough, using a variety of datasets to validate the approach more reliably. The results clearly show the superiority of this framework over all benchmarks.



Clarity:

Well-organized and written, but the complexity of the content sometimes hampers clarity, especially in dense theoretical sections.



Originality:

The integration of sequence embeddings with semantic vector quantization to handle biased generation and noisy interactions is innovative. This approach potentially offers a new perspective on improving denoising models, particularly in handling sparsity and noise within generated sequences.



Significance:

The proposed model has application in improving the quality and robustness of generated sequences, which can impact domains like recommendation systems.



Pros:

• The related works are well-explained and clearly presented.

• The main challenges and the author’s contributions are clearly outlined, with solutions effectively introduced in the introduction.

• The paper presents an innovative approach that combines vector quantization with contrastive discrepancy maximization.

• The methodology is technically rigorous and well-structured.

• Diagrams are helpful and clearly illustrate the methodology and experiments.

• Key terms are introduced clearly and sufficiently.

• Baselines are well-categorized, and all three research questions are answered clearly.




Cons:

• The paper claims to address the issue of heterogeneous user interaction sequences in recommendation tasks, which vary in length and content and contain noise due to random user behaviors. However, no experiments are demonstrating how DiQDiff handles such heterogeneous sequences, which would strengthen the paper's claims.

• In Section 4.2, the authors state that for sparse sequences with limited interactions, the closest code would provide extra information that aligns with the user’s interests, while for noisy sequences, the extracted code would enhance meaningful patterns and reduce irrelevant noise, thus improving guidance expressiveness. Instead of simply stating this, it would be more effective if this core concept, which is a main goal of the paper, was shown through mathematical formulations. Specifically, it would be helpful to see if the model maintains robustness with highly sparse sequences or if there are limitations in handling extreme sparsity, ideally demonstrated through experiments.

• In Section 4.3, the authors propose to maximize the difference between predicted item representations from different sequences using contrastive loss to prevent biased item generation. However, the scientific reasoning behind this approach is not clearly explained.

• The paper lacks a discussion of potential limitations or failure cases where this model might face challenges compared to traditional methods. Notable limitations that affect the model’s performance should be addressed.

• Although the authors compare the proposed framework's performance with baseline models, including an analysis of its computational efficiency relative to other models would enhance the quality of the paper.




Overall:

The paper introduces a strong and creative method to improve denoising in sequence generation using vector quantization and contrastive discrepancy maximization. Although it is complex and sometimes hard to follow, the work is technically solid and could be valuable in fields that use sequence generation models. Making the content easier to understand and adding more comparisons with existing methods could increase its impact even further.

**Questions:**

Please refer to the cons above.

**Reviewer Confidence:**

3: The reviewer is confident but not certain that the evaluation is correct

**Scope:**

4: The work is relevant to the Web and to the track, and is of broad interest to the community

---

### Official Review · Reviewer_tG8v · 2024-11-21

**Novelty:** 5
**Technical Quality:** 4

**Review:**

Summary:
This paper introduces Distinguished Quantized Guidance for Diffusion-based Sequence Recommendation (DiQDiff), a method designed to address two primary challenges in sequential recommender systems: the heterogeneity of user behavior and the popularity bias in recommendation models. Diffusion models (DMs) have shown promise in sequential recommendation tasks, but they face difficulties in dealing with noisy, stochastic sequences and generating diverse recommendations that meet users' personalized needs. The proposed DiQDiff approach aims to refine the guidance provided to the diffusion model by extracting more robust and effective representations of user interests, thus enhancing the model’s ability to generate more accurate and personalized recommendations.

Pros:
Novel Approach: The method introduces a distinctive solution by combining diffusion models with quantized guidance, offering a new way to address existing problems in sequence-based recommendations.
Effective in Recognizing Challenges: The paper effectively identifies key challenges in recommender systems, such as heterogeneity in user behavior and the tendency of models to over-recommend popular items.

Cons:
Lack of explanation of key concepts: The paper mentions that DiQDiff is the first method to study the combination of guided vector quantization and discriminative generation in a diffusion model for sequential recommendation. However, there is little explanation of vector quantization, which may make it difficult for readers who are not familiar with this concept to understand its role in the proposed method. The related work section only provides a brief reference to it, and it is not defined in the preliminary section, which makes it difficult for readers to grasp the technical basis of the method.

Questions about the method: The article mentions that SVQ may introduce bias because the codebook itself may limit the utilization of the precise signal, and CDM, fortunately, solves this bias problem by forcing the model to learn different patterns. However, the quantization of SVQ itself has introduced bias, why the differentiation project brought by CDM can reduce the bias introduced by quantization.

**Questions:**

Why the differentiation project brought by CDM can reduce the bias introduced by SVQ？

**Reviewer Confidence:**

3: The reviewer is confident but not certain that the evaluation is correct

**Scope:**

3: The work is somewhat relevant to the Web and to the track, and is of narrow interest to a sub-community

---

### Official Review · Reviewer_CqBN · 2024-12-01

**Novelty:** 4
**Technical Quality:** 3

**Review:**

Summary:

The paper proposes a framework to improve diffusion models (DMs) for sequential recommendation tasks. The paper mainly identifies two critical challenges:
1. Heterogeneous and Noisy Guidance: User interaction sequences often vary significantly in length and content, which can introduce noise and reduce the accuracy of user interest modeling.
2. Biased Generation: Diffusion models tend to overfit to popular items in the dataset, failing to meet personalized user needs.

To address these challenges, the authors propose DiQDiff, which incorporates two novel components:
1. Semantic Vector Quantization (SVQ): This module quantizes user sequences into semantic vectors to enrich guidance signals and better capture user interests, even with noisy or sparse data.
2. Contrastive Discrepancy Maximization (CDM): This component maximizes the discrepancy between denoising trajectories of different user sequences, mitigating biases in item generation and ensuring personalized recommendations.

The framework is evaluated on four widely-used datasets (ML-1M, Amazon Beauty, Amazon Toys, and Steam) against multiple baseline models. Extensive experiments demonstrate that DiQDiff outperforms state-of-the-art methods, achieving significant improvements in hit rate and NDCG metrics across datasets.

Pros:
1. The paper is well-organized and presented in a clear and logical manner, making it easy to follow.
2. The authors employ a fully end-to-end training pipeline that integrates SVQ and CDM seamlessly, improving the framework’s usability and scalability for real-world applications.
3. The framework is rigorously tested on four benchmark datasets, with results consistently outperforming strong baselines, including both traditional and generative recommender systems.

Cons:
1. My primary concern is that the paper’s motivations are not sufficiently reflected in the experiments. I will provide a detailed explanation in the next section.
2. The paper does not include a comparison of training and inference time overheads against the baseline methods.

**Questions:**

My main concerns lie in the fact that the paper’s motivations are not adequately reflected in the experiments. Specifically:
1. In Lines 94–95, the authors claim that “user interaction sequences in recommendation tasks are typically heterogeneous in lengths and contents.” While Table 2 provides the average lengths of user interaction sequences for the four datasets, the variance in sequence lengths is not reported. Furthermore, no experiments are conducted to demonstrate that the proposed method significantly outperforms baseline methods under varying user interaction sequence lengths. The same issue applies to the heterogeneity of user interaction sequence contents, which is mentioned but not supported by corresponding experiments.
2. In Lines 110–112, the authors state that “the denoising process that generates items is prone to mode collapse and similar generation issues, especially when biases occur in input data.” However, the cited references supporting this claim do not pertain to the recommendation system domain. As a result, I believe it is necessary to experimentally validate whether such phenomena actually occur in the context of recommendation systems. If these issues are indeed present, further experimental evidence is needed to demonstrate whether the proposed method effectively alleviates them.

**Reviewer Confidence:**

4: The reviewer is certain that the evaluation is correct and very familiar with the relevant literature

**Scope:**

4: The work is relevant to the Web and to the track, and is of broad interest to the community

---

### Official Review · Reviewer_BwKG · 2024-12-02

**Novelty:** 5
**Technical Quality:** 5

**Review:**

Pros:

-This paper introduces a generative recommendation approach, which effectively complements existing sequential recommendations.

-The paper proposes the DiQDiff method to address common data bias and diversity issues in generative recommendations.

-Extensive experiments validate the effectiveness of the method.


Cons:

-The method has significant limitations, as it is only applicable to generative recommendations.

-The experiments do not compare with classic sequential recommendation baselines such as DIN.

**Questions:**

See Cons above.

**Reviewer Confidence:**

4: The reviewer is certain that the evaluation is correct and very familiar with the relevant literature

**Scope:**

4: The work is relevant to the Web and to the track, and is of broad interest to the community

---

### Official Review · Reviewer_nNmX · 2024-12-02

**Novelty:** 5
**Technical Quality:** 4

**Review:**

This paper proposes Distinguished Quantized Guidance for Diffusion-based Sequence Recommendation (DiQDiff), addressing two key challenges in diffusion model-based sequential recommendation: heterogeneity and noise in sequence guidance, as well as bias in the generation process. Through Semantic Vector Quantization (SVQ) and Contrastive Difference Maximization (CDM), the method innovatively combines diffusion models with vector quantization techniques. Comprehensive experiments conducted on four public datasets demonstrate that the proposed approach outperforms multiple existing models in recommendation performance.

Pros:
1. The proposed DIQDIFF model effectively addresses the challenges of heterogeneous and noisy guidance, as well as bias in generation, within diffusion-based sequential recommendation.
2. This is the first work to combine semantic vector quantization with diffusion models for sequential recommendation, achieving significant performance improvements.
3. Extensive experiments conducted on four public datasets demonstrate the model's superiority.

Cons:
1. While the paper mentions that "user interaction sequences may contain noise or sparsity," the experiments do not artificially introduce additional noise or sparsity to evaluate the model's robustness. This might limit the interpretability of the results.
2. The current evaluation is limited to performance comparisons with benchmark methods, without a deeper exploration of differences in generation quality and efficiency compared to other generative models.
3. The introduction of diffusion models and semantic vector quantization may increase computational costs, yet the paper does not provide a comparative analysis of efficiency relative to other methods.

**Questions:**

1. How would the proposed model perform under artificially introduced noise or sparsity in user interaction sequences？
2. Can the authors provide an analysis of the computational efficiency of diffusion models and semantic vector quantization relative to other generative methods？

**Reviewer Confidence:**

3: The reviewer is confident but not certain that the evaluation is correct

**Scope:**

3: The work is somewhat relevant to the Web and to the track, and is of narrow interest to a sub-community